# Looking Back: Theological Reflections on the Intersection between Pentecostalism and Ubuntu within the African Section of the Apostolic Faith Mission of South Africa

Abraham Modisa Mkhondo Mzondi

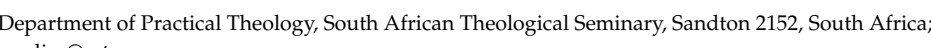

Department of Practical Theology, South African Theological Seminary, Sandton 2152, South Africa; modisa@sats.ac.za

**Abstract:** Syncretism in the African section of South African Pentecostalism followed the emergence of the Ethiopian movement. The latter took the lead in promoting the syncretising of Christianity and African culture and practice (hereinafter referred to as Ubuntu). A similar syncretism emerged in the Christian Catholic Church in Zion in Wakkerstroom, the "black section of the Apostolic Faith Mission", soon after the departure of Reverend Pieter Le Roux, who was appointed to lead the Apostolic Faith Mission in Johannesburg since John G. Lake was returning to the USA. This article intends to show that such syncretism did not occur in a vacuum. Rather, it was influenced by the interpretation of some portions of Scripture, the influence of John Alexander Dowie's praxis and some dreams and visions of a leader of the Christian Catholic Church in Zion in Wakkerstroom. This form of syncretism later permeated subsequent sections of African Pentecostalism in the Apostolic Faith Mission, resulting in the emergence of two categories of African Pentecostalism in the church: namely, those who accept this phenomenon and those who abandon it. These past developments position the Apostolic Faith Mission as a prime example to use in analysing syncretism in Pentecostalism and how it could be addressed by taking cognisance of Ubuntu without committing syncretism. Hence, the following question arises: How can theological reflections on the past experiences of the black section of the Apostolic Faith Mission of South Africa contribute to promoting a biblical approach that takes cognisance of Ubuntu without committing syncretism? This article applies the Magadi research method conceptualised for practical theology to answer this question. It further demonstrates that it is possible to promote a biblical approach that embraces Ubuntu without committing syncretism.

**Keywords:** Pentecostalism; syncretism; Ubuntu; Apostolic Faith Mission of South Africa

## 1. Introduction

Here are two perspectives on the history and syncretism of culture and religion. Kurtz (1995, p. 260) posits that "[t]he current religious scene presents a dynamic interplay between traditional practices on the one hand and widespread transplanting of traditions and experiments with syncretism on the other". Leopold and Jensen (2004, p. 5) argue that "the history of religion confirms that every religion is in 'essence' syncretic; there is no pristine origin or essence." The author holds that it is possible to hold to traditional practices without being syncretic. This view translates into Africans embracing the Christian message through the African worldview lens (Umoh 2012, p. 5) and hinges on Anderson's (1999, p. 226) view that:

> Those who censure Third-World Pentecostals for their alleged "shamanism" or "syncretism" often fail to see that parallels with ancient religions and cultures in their practices are also often continuous with the biblical revelation.

Later, Clark (2001) hinted at the challenges of syncretism in the Apostolic Faith Mission of South Africa, which is considered to be one of the largest Classical Pentecostal denominations in South Africa and is founded on William Seymour's teachings. Hence,

this article focuses on the intersection between Ubuntu and Pentecostalism in the African section of the Apostolic Faith Mission of South Africa. Zionism spread to South Africa through the efforts of the Dutch Reformed Church (DRC) minister Reverend Pieter Le Roux, who embraced John Alexander Dowie's teachings on divine healing. He was later appointed to lead the Christian Catholic Church in Zion in Wakkerstroom.

A few years later, he embraced William Seymour's teaching on Pentecostalism (baptism in the Holy Spirit with evidence of speaking in tongues) after listening to William Seymour's protégé, John G. Lake, after which he soon spread the message of Pentecostalism to the followers of Zionism in Wakkerstroom, considered the African section of the Apostolic Faith Mission (AFM). John G. Lake appointed him the leader of the AFM in Johannesburg prior to returning to the United States of America. Reverend Le Roux subsequently left the Wakkerstroom church in the hands of a local leader, Daniel Nkonyane. Nkonyane continued this work and later added some praxes he copied from Alexander Dowie (divine healing), as well as his visions and understandings of Exodus 4: 1–4 and Revelation 7: 9.

It seemed normal and natural for Daniel Nkonyane to introduce these practices in the church, since they resonate with the Ubuntu worldview. His praxis set a precedent for other members and leaders who associated with the church in Wakkerstroom and sympathised with the Ethiopian movement, which Reverend Nehemiah Tile championed in South Africa in the late 1890s. While Nehemiah Tile was the trailblazer for the Ethiopian movement that emerged in the 1890s, Daniel Nkonyane became the trailblazer for fusing Ubuntu with Pentecostalism. This development led to the emergence of African Independent Churches in the early 1900s in South Africa and in the neighbouring countries. Thus, the stage was set to syncretise Ubuntu with Pentecostalism. Contrary to Nkonyane, some Pentecostal leaders within the African section of the AFM, Elias Letwaba and Richard Ngidi, opted not to fuse Ubuntu with Pentecostalism.

The literature notes that syncretism is mixing two or more beliefs into one or adding elements to an existing belief (Mullins 2001, p. 809; Schreiter 2003, pp. 146–47). Considering that African spirituality influenced William Seymour's Pentecostalism (Togarasei 2005, p. 371), the Pentecostalism discussed in this article refers to one founded by the African William Seymour (Asamoah-Gyadu 2013, p. 143; Hollenweger 1994, p. 210). It is described as baptism in the Holy Spirit, with the evidence of speaking in tongues (Kgatle 2020b, p. 2; Martin 2004, p. 48). The question arises: How can theological reflections on the past experiences of the black section of the Apostolic Faith Mission of South Africa contribute to promoting a biblical approach that takes cognisance of Ubuntu without committing syncretism?

The Magadi research method (Mzondi 2022b, pp. 9–10, 13–14) is used to reflect on the intersection between Pentecostalism and Ubuntu within the African section of the Apostolic Faith Mission of South Africa. It is a practical theological approach that seeks to analyse a phenomenon and identify convergence points for demonstrating theological and biblical foundations to apply them to Christian practice. The method applies the three steps of **Appreciation**, **Announcing,** and **Presenting**. The first step identifies positive aspects/developments in a phenomenon, namely Ubuntu. The second step presents the common aspects between the phenomena (Ubuntu and Pentecostalism), while the third step determines if the aspects in the second step are theologically sound and biblical.

## 2. Appreciating Ubuntu

The Appreciation of Ubuntu is the first step of Magadi (Mzondi 2022b, pp. 10, 14). It discusses and appreciates the positive aspects of Ubuntu to identify the point of convergence between Ubuntu and Pentecostalism, which is the focus of the second step of Magadi (Mzondi 2022b, p. 10), **Announcing**.

*Expressions of a Supreme Being in Ubuntu*

Ubuntu promotes the existence of a supreme being. This being is believed to be the creator of all creation, including the universe, human life, animal life, and vegetation (see Mbiti 1991, p. 70). Adherents of Ubuntu in South Africa call this creator *Ramasedi* (Setswana/Sesotho), *UMvelinqangi* (isiZulu), or *Qhamatha* (isiXhosa). This creator, considered remote from humanity, can only be approached through the mediation of indigenous healers, diviners, and ancestors, who are believed to be near the creator (cf. Mbiti 1976, pp. 75–85; Nyirongo 1997, p. 51). The gap between the creator and human beings is reflected by a spiritual hierarchy that shows the position of the creator at the top, ancestors in the middle, and human beings at the bottom (Mbigi 1997, p. 54).

Adherents of Ubuntu believe that daily activities and events are influenced by the creator through the ancestors. The latter individuals communicate with humans through dreams, bad and good events, and illnesses, and humans in turn consult indigenous healers and diviners for the interpretations of dreams, bad events, or illnesses (Ntlha 2017, pp. 6–7, 30–32). The creator gives indigenous healers and diviners the ability to identify the cause of sickness/illness or bad luck and to prescribe relevant herbs to heal sickness or illness or remove the cause of bad luck. In other cases, sacrifice is the most relevant solution. These individuals may also see a person who comes to consult about their dreams or visions and tell them what to do (Hirst 2005, p. 1; Mzondi 2019, pp. 48–54, 69).

Hence, adherents of Ubuntu mainly use animal sacrifices to appease ancestors who present their request to the creator. Additionally, the practice of songs, accompanied by indigenous music instruments, and dance form an intrinsic practice among the followers of Ubuntu in ceremonies, celebrations, and rituals. Recitation of *sereto/isiduku* (reciting of clan lineage) (Mzondi 2022a, p. 2) also forms part of ceremonies, celebrations, and rituals.

As can be seen from the above discussion, Ubuntu is theistic in that it first promotes and elevates the supreme being that affects the physical. Second, it teaches that this supreme being is accessed through the mediation of indigenous healers, diviners, and ancestors. The latter communicates with human beings, including indigenous healers and diviners, through dreams, visions, incidents, and sickness/illness. Sacrifices are most appropriate to appease the ancestors, and song and dance form an intrinsic component of ceremonies, celebrations, and rituals.

## 3. Announcing Convergence and Divergence between Pentecostalism and Ubuntu

This section focuses on Magadi's second step (Mzondi 2022b, pp. 10, 14), **Announcing**. The goal of this step is to establish points of convergence and divergence between the Azusa Street phenomenon and Acts 2 and between AFM South African Pentecostalism and Ubuntu.

### 3.1. Acts 2 and the Azusa Street Pentecost Experiences

The literature points out that William Seymour and others who experienced baptism in the Holy Spirit with the evidence of speaking in tongues believed that they experienced what the Apostles and others experienced in Acts 2: 1–13 (Apostolic Faith 1906, p. 2; Cox 1995, p. 37). They also believed that other believers, both their contemporaries and future believers, could also experience it. This experience includes noises heard when singing and dancing (Mzondi 2019, p. 50). Acts 2: 4–6 describes the noise accompanied by worship and speaking in languages others heard and understood. Nel (2019, p. 2) indicates that the 1906 Azusa Street event shows that indigenous African elements had already been incorporated into Protestant Christian worship, such as trance, ecstasy, visions, dreams, and healings in continuity with similar biblical practices.

### 3.2. Ubuntu in the AFM South African Pentecostalism

Several African Pentecostals (Elias Mahlangu, Edward Motaung, Christina Nku, Daniel Nkonyane, Ignatius Lekganyane) emerged from the African section of the Apostolic Faith Mission. However, this sub-section of the article looks at how four African AFM

leaders, namely Daniel Nkonyane, Christina Nku, Elias Letwaba, and Richard Ngidi, either syncretised or opted not to syncretise Pentecostalism with Ubuntu. The first three are first-generation African Pentecostals while Ngidi is a second-generation African Pentecostal.

Mzondi (2019, pp. 48–49) points out that Daniel Nkonyane's ministry revolved around dreams, visions, worship practices, and divine healing and deliverance. Nkonyane based his teaching and praxis of barefoot worship and the use of a stick (*isikhali*/weapon) on the passage in Exodus 3: 1–6 and 4: 1–4, respectively, and he based his praxis of using a church uniform on the example of Alexander Dowie, the passages in Revelation 7: 9, and the angel in the white robe (cf. Draper 2016, pp. 58–59; Sundkler 1976, p. 53), and his dream of the white robe (Mzondi 2019, p. 48).

Nkonyane's practice of divine healing flowed from his working relation with Reverend Le Roux at the Christian Catholic Church in Zion in Wakkerstroom (Sundkler 1961, p. 48). He later left the church to establish the Christian Catholic Apostolic Holy Spirit Church in Zion. The church embraced Pentecostalism and added the praxis of African song and dance in its liturgy; this is the hallmark of Zionism in the country and neighbouring states.

Christina Nku (commonly known as MaNku) was raised in the Dutch Reformed Church and later became an ordinary member of the AFM. She experienced a series of dreams and visions from an early age. In one of them, she chose a hymn book an angel held out to her instead of money the devil advised her to choose (Landman 2006, p. 8; Mzondi 2019, pp. 74–75). She also experienced seizures and fainted frequently. Elias Nkadimeng of AFM prayed for her recovery and advised her to sacrifice a cow in honour of her ancestors as her illness was due to her divine calling (Landman 2006, p. 11). However, her dreams and visions created a rift between her and Reverend Le Roux, then the leader of the AFM of South Africa, who did not approve of them. She later left the AFM and established St. John Apostolic Faith Mission.

Later, MaNku practiced divine healing and deliverance based on her dreams and visions, where she saw her ancestors escorting her from hell and a later vision where she was instructed to use the colours blue and white for a church uniform (Landman 2006, p. 12). She later included the use of holy water in her ministry of divine healing and deliverance (Landman 2006, p. 16; Mzondi 2019, p. 76). It is not surprising that Nku's church also embraced Pentecostalism and included ancestor veneration in her church liturgy. St John Apostolic Faith Mission became the hall mark of Apostolic churches in the country and in the neighbouring states.

Consequently, Daniel Nkonyane and Christina Nku ended up including the aspect of veneration of ancestors in their liturgy because they argued that it represents their African culture and identity of dreams and visions from the ancestors, as well as received instructions related to divine healing and deliverance. The praxis of Daniel Nkonyane and Christina Nku show the intersection of the Gospel with African cultures (cf. Daneel 1987, p. 26). The pair also believed that the gift and practice of prophecy flow from the Holy Spirit, although this differed from AFM practice. Christina Nku's church describe themselves as *kereke ya moea* (spirit led church) to emphasise the role of the Spirit and prophecy in the church. The church also incorporated a unique genre of African music into their worship, resulting in unique songs, instruments, and dances that survive to this day. Nku's church sing *Sepostolo*, characterised by hand clapping, slow or fast dancing around in circles, and spinning to the beat of a drum.

On the other hand, Nkonyane's church believed in the practice of speaking in tongues and prophesying as the Spirit leads. Nkonyane's church sang what is called *isiZioni*, characterised by women singing slowly and men joining while holding sticks and moving bodies forward and backward. Dancing around and spinning in a circle also occurs.

Notably, Anderson (2000, p. 37) mentions that the Holy Spirit was central in Nkonyane's church and Nku's church, similar to the AFM of South Africa. He further classifies them as Zion-Apostolic Churches (Anderson 2000, pp. 105, 260). However, while the historic and spiritual character of these churches is consistent with their relationship to the South African AFM, Larbi (2002, p. 151) argues that they are not truly Pentecostal.

On the contrary, two Pentecostals from the African section of the AFM, Elias Letwaba and Richard Ngidi, followed a different praxis. Elias Letwaba, Nku and Nkonyane's contemporary, was raised in the Bapedi Lutheran Church (Morton 2016, p. 3), and later embraced Pentecostalism after meeting and listening to John G. Lake preaching in Johannesburg and later returned to his hometown (Burger and Nel 2008, p. 207; Kgatle 2016, p. 328). He was gifted in divine healing and is considered the first influential African Pentecostal in the AFM (Kgatle 2016, pp. 330–31; Mzondi 2019, p. 71). Letwaba's ministry shows that even though he practiced Ubuntu and embraced Pentecostalism, he opted not to fuse Pentecostalism with the Ubuntu praxis of ancestor veneration (Mzondi 2019, p. 70). Nor did he deviate from the established AFM 's liturgy, but rather followed the Pentecostal teachings he had received from John G. Lake (Kgatle 2016, p. 329; Mzondi 2019).

Richard Ngidi, originally from the American Missionary Society, later joined the AFM (Khathide 2010, pp. 43–44). He was also gifted in divine healing and saw numerous visions during his ministry (Khathide 2010, pp. 63–82).

## 4. Presenting a Suitable Approach

This section focuses on the third step of Magadi (Mzondi 2022b, pp. 10, 14), **Presenting**. It intends to present a theological and biblical approach to discerning and testing the spirit behind spiritual encounters, as stated in three passages.

Acts 16: 16–18 relates Paul's encounter with the prophesying young girl at Ephesus; I Corinthians 14: 26–33 provides guidelines for prophesying in the local house church; and I Thessalonians 5: 19–20 contains Paul's instruction to the believers not to stifle the spirit and prophecy, but to test all prophecy. These three Scriptures form hermeneutical tools to engage prophecy, dreams, and visions to propose the intended approach.

### 4.1. Prophecy, Dreams, Visions, and Divine Healing

Prophecy and the Holy Spirit were central to the practice of Daniel Nkonyane and Christina Nku. As mentioned above, Mzondi (2019, pp. 47–48, 76–77) mentioned that both experienced dreams and visions that led them to introduce spiritual aspects that resonate with the Ubuntu worldview of using tangible elements in healing like water, robes, sticks, and other objects (see Burger and Nel 2008, pp. 242–49; De Wet 1989, pp. 107, 112; Molobi 2008, p. 7).

Similarly, prophecy and the Holy Spirit are central to the AFM, which is founded on the teachings of Alexander Dowie (Oosthuizen 1987, p. 21) and John G. Lake. The latter was a former member of Dowie's church before joining William Seymour (Burger and Nel 2008, pp. 31, 91, 96; Clark 2009, pp. 175–76). The church also denounces the use of alcohol, polygamy, and the use of tangible elements in the liturgy (Larbi 2002, p. 148). Dowie and Pentecostalism hold that the Holy Spirit is the enabler of visions, dreams, and divine healing as the fulfilment of the Lucan corpus (Luke 24: 49; Acts 1: 8; 2: 1–6). The AFM prides itself on being one of the denominations that hold and believe that this phenomenon will not stop happening (Nel 2008).

#### 4.1.1. Applying the Scriptures to Address the above Praxis

Some AFM theologians confirmed the use of tangible elements in the African Pentecostal section (Burger and Nel 2008, pp. 242–49; De Wet 1989, pp. 107, 112; Molobi 2008, p. 7). Hence, the author opts to apply Figure 1 below, adapted from Mzondi (2010, p. 139), to demonstrate the sequence of discerning and testing the spiritual praxis in the context of the AFM and the wider Pentecostal Christian tradition.

The first step is testing the practice and the role of visions, dreams, and divine healing in the ministry of Daniel Nkonyane and Christina Nku. The next step is testing Nkonyane's interpretation of Exodus 4: 1–4 and Revelation 7: 9. The last step is testing the gift of divine healing in the ministries of Nkonyane, Nku, Letwaba, and Ngidi.

Figure 1 promotes moving from the text to the context by using texts that demonstrate the use of visions, dreams, prophecy, and healing.

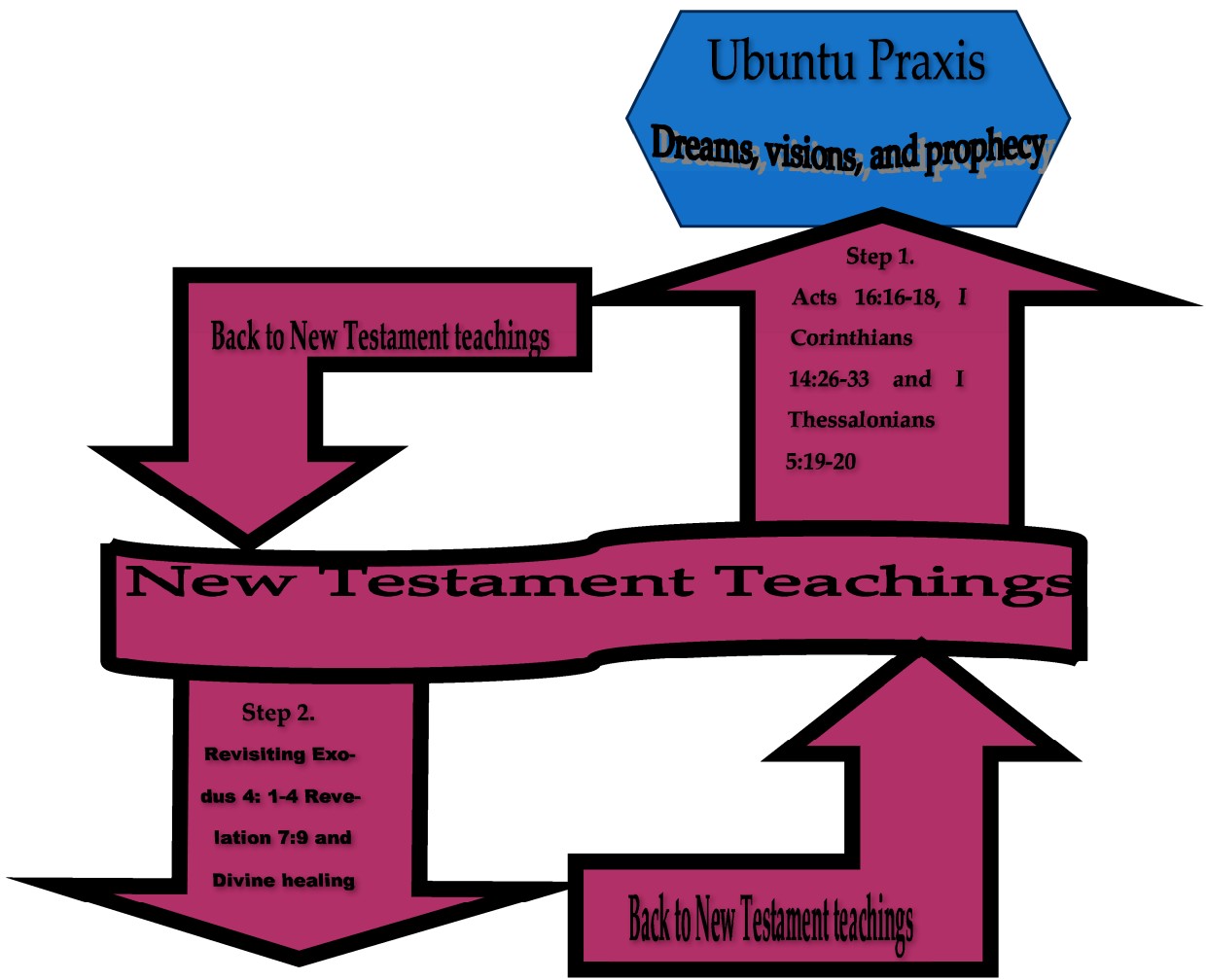

**Figure 1.** Movement from New Testament teaching to and from *Ubuntu* praxes of dreams, visions, divine healing, and Bible interpretation.

Two steps flow from Figure 1 above: namely discerning prophecy, dreams, and visions, and revisiting Nkonyane's interpretation of Exodus 4: 1–4 and Revelation 7: 9 as well as his divine healing praxis. The first step assists in distinguishing between prophecy and divination (Kgatle 2019, pp. 3–4). The second step helps to judge the teachings of Pentecostal pastors (Keener 2014, p. 375). The two steps flow from the view that Pentecostals share a high view of Scripture and use the Protestant Bible (Nel 2020, p. 1). Figure 1 echoes Shingange's views about applying Africa Biblical Hermeneutics (ABH), as he argues that African Pentecostals should "return to the dependency on the Bible; however, they should also seek to strike a balance between the importance of the Word of God, the need for Spirit, and the African realities. This will mean the contextualisation of reading the Bible in the light of the primacy of the Spirit whilst taking cognisance of the unique needs and realities of African people". (Shingange 2021, p. 34).

4.1.2. Step 1: Discerning Prophecy, Dreams, and Visions

Paul is the relevant New Testament model of discerning prophecy, dreams, and visions. As noticed above, Acts 16: 16–18, I Corinthians 14: 26–33, and I Thessalonians 5: 19–21 are considered to apply the process in Figure 1.

Acts 16: 16–18

Paul's encounters with the slave girl in Acts 16: 16–18 occurs in the context of his third missionary journey guided by the Holy Spirit (Holladay 2017, p. 370) as he saw a

vision of a man asking him to come to Macedonia. He and his team went to Macedonia and reached the capital city, Philippi, where they spent some days with Lydia, who believed in the Lord (Acts 16: 9–15), and later encountered a young girl prophesying about them (Acts 16: 16–17). His encounter with the girl on the way to a prayer meeting suggests three steps, namely exercising quietness, listening to the Holy Spirit, and taking the appropriate actions. Hearing the girl prophesy (Acts 16: 17) for many days (Acts 16: 18) did not overwhelm him; instead, he remained unmoved. His posture suggests that he was listening and seeking divine direction or intervention (Acts 16: 18). Hence, he was troubled, seemingly after ascertaining the source of the prophecy, causing him to act appropriately (Acts 16: 18). This approach teaches Pentecostals not to be quick to act but to listen to the Holy Spirit, who in turn will guide and cause them to take appropriate actions. Paul's action teaches that spiritual sources, other than the Holy Spirit, can give correct prophecy.

Applying Paul's approach suggests that Pentecostals should engage prophecy, dreams, and visions by first moving from the Scriptures, then moving to the context (dreams and visions), as other spiritual sources can provide correct prophecy. Second, they should listen to the Holy Spirit for guidance on taking appropriate actions. Third, they should take the appropriate actions after hearing from the Holy Spirit.

I Corinthians 14: 26–33 and I Thessalonians 5: 19–21

The Corinthians verses covers the teachings on gifts of the Holy Spirit and how they should be operated in the local church (Carson et al. 1992, p. 261), while the Thessalonians verses contains a warning that believers should live appropriately (Carson et al. 1992, p. 357). Paul advised the believers in Corinth and Thessalonica to allow the gift of prophecy to function under specific instructions. First, believers should not despise the gift of prophecy but allow its praxis (I Cor 14: 30; I Thess 5: 19; Hamp 2017, p. 228). Prophecy should edify believers and not cause confusion (I Cor 14: 26, 33). Second, this should be performed by allowing three, one person at a time, to prophesy while others discern what is said as they exercise spiritual restraint (I Cor 14: 29, 32; Keener 2005, pp. 117, 119). Third, believers should test every prophecy (I Thess 5: 21). Paul's instructions in the two Scriptures suggest that Pentecostal local churches should first have a team of prophets who will discern each other's prophecies and test the spirit behind each prophecy. Only three people should prophesy, one at a time. Second, prophecy should edify the church.

4.1.3. Step 2: Revisiting Nkonyane's Interpretation of the Scripture and Practice of Divine Healing

Exodus 4: 1–4 and Revelation 7: 9

Exodus 4: 1–4 is found in the context of God appearing to Moses and instructing him to deliver the nation of Israel, and Moses' concern that they will not believe God sent him (Hammilton 2011, pp. 69–70). Revelation 7:9 appears in the interlude that shows the 144000 saints between the sixth and seventh seals (Holladay 2017, p. 852).

Daniel Nkonyane used Moses' encounter with God of the burning bush in Exodus 4: 1–4 to introduce the practice of using a stick in divine healing and deliverance. He also followed Michael Mngumezulu's understanding of the apostle John seeing many people from different nations dressed in white robes (Rev 7: 9) and accepted Alexander Dowie's practice of wearing a white gown (see Sundkler 1976, pp. 48–50).

Nkonyane's understanding of the Scriptures resulted in him introducing the use of tangible elements in the church's liturgy. The process of placing the two Scriptures in Figure 1 shows that Nkonyane understanding of the texts was literal and experiential (Nel 2021, pp. 3–4). Hence, Pentecostals are often accused of the same tendency seen in current arguments about the subjective interpretation of the Bible (Nel 2017; Resane 2017) that leads to incorrect and dangerous practices (Keener 2016, pp. 107–8; Kgatle and Anderson 2020).

This error can be resolved by applying the practice of the Berean church (Acts 17: 10–11) as a relevant model to encourage local Pentecostal church members to engage the teachings of Pentecostal pastors so that they can differentiate between true and false teachings. The

practice of the Berean church points to a search for consistency with the Scriptures, like the Jewish practice of examining the Scriptures and listening diligently (Keener 2014, p. 375). Thus, providing the necessary checks and balances to avoid subjectivism in biblical interpretation and promoting the reading of the Bible in the community of believers recognises the Pentecostal approach from text to experience and back to the text (Grey 2011, p. 154).

Divine Healing

Pentecostals hold that the gift of healing is one of the continuing spiritual gifts of the Holy Spirit (1 Corinthians 12: 9; Nel 2008, 2021, pp. 3–4). Nkonyane, Nku, Letwaba, and Ngidi have all practiced divine healing in previous generations of African Pentecostalism (Anderson 1992; Landman 2006; Kgatle 2020a; Khathide 2010).

However, Nkonyane and Nku advocated for the use of tangible elements in divine healing (Anderson 199, pp. 107–8), while Letwaba and Ngidi did not (Kgatle 2017, pp. 6–7; Kgatle 2020a, pp. 8–9; Mzondi 2019, p. 83). Letwaba and Ngidi imitate the miracle recorded in Acts 3: 1–11 (the healing of the cripple at the Temple), where Peter and John healed him in the name of Jesus.

The current African generation in the AFM continues to practice healing by the laying of hands in the name of Jesus. Applying four of the five *solas* of the Reformation (Strauss 2021, p. 1), *sola scriptura, sola gratia, sola fide, and soli Deo gloria*, it is suggested that Pentecostals from the AFM and other Pentecostal churches should note that the gift of healing flows from the grace (*sola gratia*) of the Holy Spirit. Therefore, it should be used in the light of the Scripture (*sola scriptura*), using the name of Christ alone and faith alone (*sola fide*), and aiming at the glory of God alone (*soli Deo gloria*).

## 5. Conclusions

The focus of this article was to provide some theological reflections on the past experiences of the black section of the Apostolic Faith Mission of South Africa. The reflection contributes to promoting a biblical approach that takes cognisance of Ubuntu without committing syncretism. This was seen in the 1906 Azusa Street event that involved the manifestation of some indigenous African aspects in continuity with similar biblical practices in Acts, thus promoting the view of embracing Ubuntu without committing syncretism. It uses the Magadi practical theology research approach to, first, appreciate Ubuntu; second, show points of convergence between Ubuntu and Pentecostalism; and third, propose a biblical and theological approach that promotes the practice of prophecy, visions, dreams, sound interpretation of the Scripture, and praxis of divine healing among Pentecostals in the Apostolic Faith Mission of SA and other Pentecostal churches.

**Funding:** The research received no external funding. The APC was funded by SATS.

**Institutional Review Board Statement:** Not applicable.

**Informed Consent Statement:** Not applicable.

**Data Availability Statement:** Not applicable.

**Conflicts of Interest:** The author declares no conflict of interest.

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
