# Peer review of "Looking Back: Theological Reflections on the Intersection between Pentecostalism and Ubuntu within the African Section of the Apostolic Faith Mission of South Africa"

_religions, doi:10.3390/rel14101274_

Round 1
Reviewer 1 Report
In terms of contents, the present article addresses several important issues and makes a significant contribution to understanding the roots as well as the historical developments of Pentecostalism in (South) Africa in general and in the black section of the Apostolic Faith Mission of South Africa in particular. When it comes to formal and methodological issues, though, the paper needs to be restructured and revised.
The first methodological issue concerns the disciplinary perspective of the study. While the paper’s title suggests a historical approach (Church History), its aim conveys the impression that it is about Bible Studies. Throughout the text, a few pastoral and theological issues are dealt with, too. Even though the study can have an interdisciplinary character, designing the investigation within the framework of a specific discipline may help keep both the focus and a specific method.
The second—and most important—aspect is relative to the question of syncretism. In the text, the term is dealt with in an ambivalent way. At the very outset, two understandings of religion are presented, with each suggesting a different interplay between religion and syncretism. However, the author(s) do(es) not clearly take up any position: whether any religion is, in principle, syncretistic or not. Therefore, I’d recommend taking up a position in this regard and justifying it, in the first place. Moreover, if the author(s) wish(es) to make a case that the black section of the Apostolic Faith Mission of South Africa has not been a syncretistic instantiation of Pentecostalism, a precise definition of “pure Pentecostalism” should be provided as well. To that end, it does not suffice, in my view, to state that “The literature points that William Seymour and others who experienced baptism in the Holy Spirit with the evidence of speaking in tongues believed that they experienced what the Apostles and others experienced in Acts 2:1–13 (lines 120–22).” It is not sufficient, because (a) it is too vague and (b) not even “the literature” about it is provided. Accordingly, the author(s) should provide evidence why “pure Pentecostalism” does include “the practice of prophecy, visions, dreams, sound interpretation of the Scripture and praxis of divine healing (lines 367-68)” while other forms of Pentecostalism do not.
This leads me to my third remark. If the black section of the Apostolic Faith Mission of South Africa is considered an instantiation of “pure Pentecostalism”, as against syncretistic forms, it cannot be drawn as a parameter for developing a biblical approach. And neither can Ubuntu, in my view. Therefore, I cannot see yet the point of “taking cognisance of Ubuntu” for developing a biblical approach. Of a theological study, I would expect a theological justification why the aforementioned practices are consistent with divine revelation—and hence with the Scriptures—and not just the statement that these practices are found both in the scriptures and in Ubuntu or the black section of the Apostolic Faith Mission of South Africa. In other words, what is the theological significance or the biblical status of these practices? What do they reveal about God? Historical examples of “biblical truths” are but instantiations of it. As such, they do not replace theological work.
Finally, the option for the Magadi research method (Mzondi 2022b, pp. 9-10,13-14) should be reconsidered, in my view. It may certainly be useful “to reflect on the intersection between Pentecostalism and Ubuntu (lines 71-72)”, but not for developing a biblical approach, in the light of what has been expounded above. If the method is indeed appropriate, it should be better explained. The few lines used to justify this choice do not suffice, in my opinion. For it falls short not only of a biblical criteriology, but also of a theological one. For example, when it comes to the second step of the method, namely appreciation (line 80), on the basis of which (theological) criteria can one “appreciate Ubuntu” or define “the positive aspects of Ubuntu”?
In addition to these remarks, I’d recommend having the article proofread. Please, consider revising the following spelling or grammar mistakes, while being aware that the text may contain more mistakes.
Line 4: Apostolic Fath Mission
Line 6: emergence of. the
Lines 15-16: those who embraced this phenomenon, and those who renounced the phenomenon.
Line 40: Hence, this chapter ::: Is it a chapter?
Line 41: check indentation
Line 47: prodigè
Line 46: check indentation
Line 56: set a precedence <–> set a precedent
Line 66: syncretism is mixing two or beliefs
Lines 168-69: the Holy Spirit enable the gift
Lines 217-18 (Burger and Nel 200, pp. 31 …)
Line 356: sola scriptura, sola gratis, sola fide and soi Deo gloria
Line 358: sola gratis
Finally, please, revise the formatting of the text. Issues such as: Interline spacing (e.g. lines 67-70, 75-77), text justification (e.g. lines 66-70, 71-77), Font size disparities (273-276), etc. Also, the graphic (lines 241-264) takes up, in my view, too much room in the article. It can certainly be smaller and perhaps also aesthetically optimized.
Reviewer 2 Report
Congratulations to the author/s for a well-written, well-structured, and adequately referenced article. Your work follows logic in the manner in which you present your argument. Again, the article presents and follows a well-articulated methodology. Indeed, this paper contributes to the body of knowledge within the African Pentecostal Scholarship.
Kindly check and remove the full stop that has been erroneously added to the second line of the abstract.
1. The research addresses the intersection between Pentecostalism and Ubuntu within the African section of the Apostolic Faith Mission of South Africa without being syncretic. 2. The topic is original in the manner in which the concept of Ubuntu has been interpreted and used to show its relation to African Pentecostalism and syncretism. It does address the usage of Ubuntu which in most cases is not integrated into syncretism. 3. It adds the element that projects Ubuntu as compatible with African Pentecostalism/ Spiritualities, rather than discrediting this concept, the article demonstrates its usefulness, however, without falling into the trap of syncretism. 4. The methodology has been well presented and it is easy to follow the argument in the way the methodology has been used. 5. The conclusion is consistent with the abstract, introduction, and the entire discussion in the manuscript. 6. The article has been adequately and appropriately referenced in that all the arguments made are well-substantiated. 7. The author/s stated that the figure that has been used was adapted from Mzondi (2009a, P139) this provides a disclaimer that allows the usage of the figure without resulting in plagiarism. Again, the figure has been well-discussed to contextualize its usage.
Author Response
The full stop removed.
Round 2
Reviewer 1 Report
I have gone through the revised version of the article "Looking back and forward", and I do recognize, on the one hand, the author(s)’s efforts to engage with my review/critique as well as to change, complement, and improve the quality of his/her/their text on that basis. On the other hand, my critique addressed rather methodological and structure-related issues, to which one cannot do justice by simply adding a few more sentences, as was the case.
In addition, there are still a few spelling mistakes and formatting issues that have necessarily to be addressed, including the graphic, which, at least in the version that I downloaded, looks quite strange.
Considering that, I do not think that this article can be published in its present/revised form. If not the content/method-related issues that I addressed in my review, at least the formal ones (proofreading, formatting, graphic) still need to be revised, in my opinion.
I recommend a final proofreading.
